# Space–Time Dynamics of African Swine Fever Spread in the Philippines

**DOI:** 10.3390/microorganisms11061492

**Published:** 2023-06-03

**Authors:** Chia-Hui Hsu, Maximino Montenegro, Andres Perez

**Affiliations:** 1Center for Animal Health and Food Safety, College of Veterinary Medicine, University of Minnesota, Saint Paul, MN 55108, USA; hsu00124@umn.edu; 2Pig Improvement Company (PIC) Philippines, Pasig City 1605, Philippines; max.montenegro@genusplc.com

**Keywords:** African Swine Fever, spatial analysis, spatiotemporal cluster, seasonal pattern, Philippines, Central Luzon, disease control

## Abstract

African Swine Fever (ASF) has threatened the swine industry of Southeast Asian countries, including the Philippines, since 2019. Given the severity and the economic impact of the ASF epidemic, understanding the spatial and temporal patterns of the disease is crucial for devising effective control measures. Here, data on 19,697 ASF farm outbreaks reported in the Philippines between August 2019 and July 2022 were analyzed to estimate the space–time clustering, seasonal index, and directional spread of the disease. Central Luzon was the region with the largest number of reported outbreaks, followed by Regions I and II, whereas Western and Central Visayas remained ASF-free throughout the study period. ASF outbreaks were temporally and spatially clustered and exhibited a distinct seasonal pattern, with highest and lowest frequencies reported between August and October, and April and May, respectively. This seasonal pattern may be explained, at least in part, by a combination of environmental and anthropogenic factors, such as rain and cultural practices leading to disease spread. The results here will help inform decisions intended to mitigate the impact of ASF in the Philippines and will contribute to the understanding of the epidemiological dynamics of one of the most important emerging swine diseases globally.

## 1. Introduction

African Swine Fever (ASF) is a highly contagious viral disease that affects both domestic pigs and wild boars. Clinical signs of the disease include high fever, loss of appetite, vomiting, diarrhea, and sudden death. ASF is caused by the African swine fever virus (ASFV), which belongs to the *Asfarviridae* family, and it is known to be a serious threat to the global pork industry. There is no known cure for ASF, and infected pigs must be depopulated or culled to prevent disease spread. While ASF is not harmful to humans, it is an important transboundary animal disease that spreads through direct or indirect contact with infected pigs, contaminated feed, and other pork-related materials. ASF outbreaks might impose a significant impact on the pork industry and lead to international and national trade restrictions, impacting food security, and causing substantial economic losses to affected countries.

In 2007, ASFV spread into and through the Caucasus region including Armenia, Azerbaijan, and Georgia, as well as Russia [1]. The outbreaks in the Caucasus region were thought to stem from the illegal importation of contaminated meat from East Africa, whereas the epidemic in Russia was primarily linked to wild boar movements and contact with traditional free-range backyard farms [2,3]. The disease subsequently spread eastwards and China’s first ASF outbreak was reported in the northeastern city of Liaoning province in August 2018, leading to major disruptions in the country’s pork production and causing shortages and increased prices for pork products since then. In Southeast Asia, the rapid spread of ASF has also impacted a number of countries, including Vietnam, Cambodia, Laos, Philippines, Myanmar, Timor-Leste, Indonesia, Malaysia, and Thailand, between 2019 and 2022. The threat of ASF continues to loom over the Southeast Asian pork industry.

In July 2019, the Philippines reported its first cases of ASF in Rizal province near Manila [4]. Since then, the virus has spread into other provinces, including Bulacan, Pampanga, Nueva Ecija, and Cavite. ASF outbreak significantly impacted Philippine pork industry, resulting in the culling of over 300,000 pigs. The growth rate of pork production, hence, dramatically decreased by 20.8% in 2021 [5]. Beyond affecting farmers, ASF led to higher pork prices, impacting vulnerable consumers’ livelihoods [6]. In response to the outbreak, the Philippine government has implemented various policies and public health strategies [7], such as the National Zoning and Movement (NZM) plan for African Swine Fever, which designates different zones based on ASF risk levels and restricts movements between regions.

Inspired, at least in part, by actions taken that led to the successful implementation of the National Foot and Mouth Disease Eradication Program in the Philippines [8], a set of actions referred to as the 1-7-10 protocol was conceptualized and implemented to contain ASF spread. The 1-7-10 protocol was regulated through an administrative order passed by the Department of Agriculture, providing guidance for its implementation. Key components of the 1-7-10 protocol include the rapid culling of domestic pigs within a 1 km radius of ASF-infected farms, enforcing active surveillance activities and testing within a 7 km radius, and requiring swine farms within a 10 km radius to submit a mandatory report on disease surveillance [7,9]. The stamping-out of animals within 1 km from the infected premises, referred to as the quarantine zone, should be completed in five days or less, regardless of the population.

On 10 May 2021, Presidential Proclamation No. 1143 declared a State of Calamity throughout the country for one year, acknowledging the urgency to mitigate economic losses and the need to ensure food supply [10]. The proclamation allowed local government units (LGUs) to allocate funds for rapid response initiatives, and the Armed Forces of the Philippines provided support to maintain peace and order in affected areas.

In addition, Bantay ASF sa Barangay Program (BABay ASF), a National African Swine Fever Prevention and Control Program, was launched in 2021, and, by the time when this manuscript was written in May 2023, it was still ongoing. The program aimed to prevent and control ASF through surveillance, monitoring, and repopulation efforts. It involved information dissemination, biosecurity measures, and collaboration with LGUs. The program faced challenges such as delays in translating materials to local dialects, resistance from certain stakeholders, COVID-19 protocols, and manpower shortages [11,12].

Despite ASFV has transmitted through susceptible populations of the Philippines for more than three years, key aspects of the epidemiological dynamics of disease spread in the country are yet to be elucidated.

The objective of the manuscript here was to conduct a comprehensive assessment of the spatial and temporal dynamics of the ASF epidemic in the Philippines. The study focused on understanding the spread of the ASFV in both space and time scales, identifying potential risk factors, and developing strategies to control the outbreak and enhance early detection in the future. This research provides insights into the dynamics of the ASF epidemic in the Philippines and help inform the development of effective control measures. Specifically, the results here will help Philippine stakeholders inform decisions associated with the implementation of the NZM plan and strengthening national surveillance systems, aligning with the main objectives of BABay ASF program. The findings are also of interest for stakeholders in countries other than the Philippines, to increase the body of knowledge related to the epidemiology of the disease and support the design and implementation of prevention and control plans.

## 2. Materials and Methods

### 2.1. Data

Data on farm outbreaks officially reported to the government of the Philippines between 16 August 2019 and 20 July 2022 were made available to us by the International Training Center on Pig Husbandry (ITCPH), Agricultural Training Institute, Department of Agriculture of the Philippines. The Philippines is divided into three primary geographical regions, namely, Luzon, Visayas, and Mindanao. To facilitate its administration, the country is further divided into 17 regions, 81 provinces, 145 cities, 1489 municipalities, and 42,029 barangays, as of the year of 2022. The barangay is the smallest organizational level of administration in the country. Recorded data, for each farm-outbreak, included event ID, administrative level (region, province, municipality, and barangay), and date of reporting.

According to the definition provided by the World Organization for Animal Health (WOAH), an outbreak is characterized by the identification of one or more cases within an epidemiological unit. According to national regulations, farms (backyard or commercial) are considered the epidemiological unit of reporting in the Philippines. Subsequently, outbreaks are defined as at least one ASF case identified on a farm. Outbreak location was approximated using the centroid (latitude and longitude) of the barangay in which the affected farm was located. A total of 19,742 farm-outbreaks were recorded in the raw document; however, 45 farm-outbreaks (0.23%, 45/19,742) were manually removed because of incomplete information. Hence, 19,697 farm-outbreaks were then used and analyzed after data cleaning. Data were organized in a generic database and descriptive statistics were computed using Microsoft Excel, version 2016 and R software, version 4.2.0. In addition to the descriptive statistics, for exploratory data analysis, the epidemic curve was built, and a biannual (6-month) scale was used to describe the fluctuations in the first and the second half of each year.

### 2.2. Spatial Statistical Methods

The mean center, spatial distribution, and spatial autocorrelation were computed or assessed using the ArcGIS software (Pro 3.1.0) (ESRI Inc., Redlands, CA, USA). The spatial distribution of the disease was depicted using choropleth maps. Spatial autocorrelation refers to the phenomenon of similar values tending to cluster together in space. In other words, it is a measure of the similarity between the values of a variable at different locations, and it is used to describe the overall patterns of spatial dispersion or clustering in the ASF outbreaks. Moran’s Index was used to assess the autocorrelation in the ASF outbreaks. The value of Moran’s index fluctuates between 1 and −1; positive values suggest that similar values tend to cluster together, whereas negative values suggest that disease is over dispersed, and values close to 0 are suggestive of randomness in the spatial distribution [13]. In simple terms, a cluster, as described in the following context, refers to a group or collection of outbreaks that are significantly closely located (geographically, temporally, or both) to each other.

### 2.3. Seasonal Index (ASi)

In order to investigate if there is a specific pattern in the occurrence of ASF outbreaks on farms, a seasonal index (S*i*) was calculated for each month. The S*i* was computed as the ratio between the number of ASF outbreaks reported in a given month (O*i*) and dividing it by the average cumulative number of outbreaks across all months during the study period (M).
S*i* = O*i*/M

A moving average of three months (AS*i*) was then applied to the resulting values of S*i* in order to account for random fluctuations. If the number of ASF outbreaks in a given month deviates from what would be expected, the resulting value of AS*i* will be either greater or lower than 1.

### 2.4. Space–Time Permutation Model

A methodology that combines spatial and temporal factors, known as a space–time permutation model of the scan statistic, was chosen to evaluate the clustering of ASF outbreak cases in both space and time. This approach was used as only reported cases were available, whereas reliable data on the spatial distribution of swine population at risk of ASF was not [14]. The analytical technique involves computing a cylindrical window with a circular geographic base and a height that corresponds to the time period. This window is moved across the study region and alternatively centered on each outbreak location and the expected number of cases within the window is calculated based on the assumption that the cases are randomly distributed in space. For the analysis, the spatial window was set to a maximum of 50% and the temporal window was set to a minimum of 1 month and a maximum of 4 months, which is equivalent to a season. Based on the optimal spatial and temporal parameters, space–time permutation analysis was applied to identify the geographic regions and time periods of potential clusters with significantly higher ASF incidents in comparison to surrounding areas. The analysis was performed using the SaTScan software v10.0.2 (Kulldorff, Cambridge, MA, USA).

### 2.5. Directional Analysis of Space–Time Clusters

For each space–time cluster identified, the direction of disease spread was evaluated using a directional test. The directional test was performed using ClusterSeer software (version 2.5.2) (Biomedware Inc., Ann Arbor, MI, USA), and the relative mode was chosen following the user manual book 2 version 2.5.

The null hypothesis of the directional was that there was no association between the times at which cases occur and the directions of the vectors formed by connecting the spatial locations of the case, whereas the alternative hypothesis indicated that direction from one case to the next is similar for cases that occur at approximately the same time.

The angles were measured counterclockwise from the horizontal axis, with east indicated as a value of 0°, north as 90°, west as 180°, and south as 270°. Secondary directions were described to assist in interpreting the angles. The term “concentration” in directional analysis refers to how cases align in a particular direction. The concept is analogous, although not exactly the same, to that of “precision” (the reciprocal of variance) in statistics, except that here, values range from 0 to 1, indicating total absence of, and perfect alignment of outbreaks in a given direction, respectively. The significance of the test statistic results was estimated using Monte Carlo simulation (*p* < 0.05).

### 2.6. Interpretation of Results

Results were presented to a group of 25 members of the Philippine veterinary services (*n* = 10) and swine industry (*n* = 15), at a workshop organized in Malvar, Batangas, in May 2023 by the International Training Center on Pig Husbandry (ITCPH), which is a training center of the Philippine, Department of Agriculture. Workshop participants had, on average, 20 years of experience in swine veterinary practice or service in the Philippines. Results were introduced to participants, which were asked to anonymously provide their opinion on (a) alignment of the results with their perception of the epidemiological situation of ASF in the country, and (b) factors that may be associated with the time-space dynamics observed in the results. Participants were subsequently asked to discuss their responses, initially in four tables, followed by a group discussion.

## 3. Results

The highest number of outbreaks was reported in the second semester of 2020 (*n* = 5486 outbreaks), representing 27.85% of all reported cases throughout the 3-year study period. In 2020 and 2021, the two years for which data were available for the entire year, the epidemic was more severe in the second semester compared to the first semester of the year (Figure 1). The seasonal index value (S*i*) was highest in August (S*i* = 1.79) and September (S*i* = 1.67) and lowest in May (S*i* = 0.48) (Figure 2), reinforcing the conclusion that the epidemic was most severe over the second half of the years.

Most (15/17) administrative regions of the Philippines reported ASF outbreaks within the study period. Region III, also known as Central Luzon, experienced the highest number of ASF outbreaks, accounting for 24.66% of all reported cases (4857/19,697). The Ilocos Region, also known as Region I, had the second-highest number of outbreaks, with a cumulative total of 3404, followed by Cagayan Valley in Region II, with a total of 2664 outbreaks. These figures demonstrate the significant impact of ASF on several regions in the northern part (Luzon island) of the Philippines. Region VI (Western Visayas) and Region VII (Central Visayas) maintained ASF-free zone during the study period (Figure 3).

The distribution of all reported ASF outbreaks was spatially clustered as suggested by the results of the Global Moran’s I test (Moran’s Index = 0.1186; *p* < 0.0001).

The space–time scan analysis identified five significant (*p* < 0.01) clusters (Table 1). The first three clusters (#1 to #3, Figure 4) were consecutive and located in Luzon island, whereas the fourth cluster, with the largest radius of approximately 400 km, centered in East Visayan island (Region VIII), representing potential risk for larger areas of ASFV spread between Visayan and northern Mindanao. The fifth cluster was identified in Northern Luzon island again, indicating a resurgence of the ASFV epidemic in similar pattern of space and time in this area, with a larger radius of 107 km compared to cluster #1 to cluster #3. For each geographical region, the radii of the space–time clusters increased over time.

The directional tests in five space–time clusters were all statistically significant (*p* < 0.01) and indicated different directions of ASF spread at the local level (Figure 4).

Most workshop participants (*n* = 24, 96%) indicated that the results presented here reflect, in their experience, the time-space dynamics of ASF in the Philippines. The follow-up discussion reached unanimous consensus in that regard. Reasons described to the participants to, potentially, explain the results included (1) a drop in price of swine and an increase in the number of pig sales typically observed in the third quarter of every year, associated with the need to secure funds to support costs associated with the beginning of the academic year in the Philippines, which results in an increase in the movement of pigs to markets and slaughterhouses; (2) heavy rains typically occurring during the second semester each year, which results in floods that may contribute to environmental contamination and spread of the disease.

## 4. Discussion

ASF has unprecedentedly spread globally since 2007, imposing far-reaching losses to affected countries and resulting in an extraordinary challenge for the veterinary services to prevent or mitigate the impact of the disease [15]. The present study assessed, for the first time, the epidemiological dynamics of ASF spread in the Philippines. The results presented here will help inform the implementation of control activities in the country, and increase awareness and understanding of disease spread, which is of interest for the international veterinary community.

The higher incidence of ASF outbreaks observed in the second semester of each assessed year may be attributed to various factors. Notably, the period of highest disease incidence coincides with the rainy (or monsoon) season, which typically occurs in the Philippines between June and November [16]. Furthermore, August and September, which are typically the peak months of the rainy season, had the highest seasonal index values, suggesting a higher frequency of ASF outbreaks during this period. There is a significant relationship between ASF outbreaks and the amount of precipitation [17]. These results suggests that higher humidity levels, often associated with more rainfall, may contribute to the higher occurrence of ASF outbreaks in certain times of the year. Flooding can lead to the dispersal of carcasses and the subsequent contamination, causing leaching of the ASFV into the ground. This increases the risk of virus re-emergence during subsequent floods.

The dry season in the Philippines can be further divided into the cool dry season (December to February) and the hot dry season (March to May). The ASFV is also sensitive to environmental temperature, with lower temperatures being more favorable for viral survival [18]. Based on the seasonal index results and the characteristics of the disease pattern, it can be inferred that the cool dry season between December and February, with lower temperatures and humidity levels, is less favorable for the spread of ASF. On the other hand, the hot dry season from March to May, with higher temperatures and less rainfall, could potentially result in a lower occurrence of ASF outbreaks.

The spatial autocorrelation using Moran’s I index revealed an uneven distribution of the ASF outbreak in the Philippines. In addition, the results of the space–time clusters analysis showed that the disease was concentrated in the northern part of the country, with Manila as the initial outbreak location. ASF eventually spread west- (Region I) and east- (Region II) wards through Luzon island during 2020, and north-wards in the second semester of 2021. This clustering had a significant impact on hog production in regions III (Central Luzon) and IV-A (Calabarzon), which were previously the top hog production regions with 16.16 and 12.35 percent of the country’s total swine inventory before ASF hit [19]. The space–time pattern suggests that swine producers in Luzon Island faced a greater disease burden and more significant challenges in repopulating hogs than those in other islands. In the annual swine report of 2021, it was noted that the hog inventory of the top producing regions in the Philippines has now shifted to Western Visayas and Central Visayas, with 12.1 and 11.6 percent [5], respectively, of the country’s total swine inventory. The impact of ASF and its re-occurrence in the northern island has made it challenging for swine farmers to develop sustainable long-term solutions to protect their herds and livelihoods. Targeted surveillance and biosecurity measures may be needed in these high-risk clustering areas to mitigate the impact of the disease.

The space–time ASF clustering in Luzon island may be explained, at least in part, by anthropogenic and demographic factors that facilitate disease spread. Specifically, a combination of high human and pig population density, particularly in IV-A (Calabarzon) where most of the commercial farms are located, and relatively frequent human and animal movements in this region could have contributed to the occurrence of the clusters identified in this paper. Additionally, a closely connected network of swine and swine-related products may have facilitated disease spread. In contrast, these factors are less prevalent in the southern part of the country. Despite the largest radius of cluster #4, the number of outbreaks did not correspond to a higher value. Overall, the northern part of the country had a higher impact due to the concentration of outbreaks in this region during the study period.

African Swine Fever is commonly assumed to be transmitted through pig movements and the movement of people who come into contact with infected pigs or contaminated pork products [20,21]. In China, there was a notable increase in ASF outbreaks in rural areas from December 2018 to January 2019, which coincided with the Chinese New Year and the nationwide trade of animals and pork-related products [22,23]. In Vietnam, ASF was first reported in February 2019 [24], peaked in March. The first outbreak was brought under control within seven months in 2019, but re-emerged in April 2020, potentially due to increased movement of people and animals during the Tet holiday (Vietnamese New Year) [25,26]. Veterinarians with field ASF experience in the Philippines indicated that, during the second half of the year, particularly during the third quarter coinciding with the start of academic year, smallholders tend to sell pigs fund education. However, because a reduction in price of pigs is also typically seen at this time of the year, movements in search of best prices are common throughout the country. The increased human activity can result in more frequent movement of animals and pork-related products, which may contribute to the spread of ASF. As a result, the seasonal pattern observed in our analysis in the Philippines may also be explained, at least in part, by this social phenomenon.

Limitations of this study are mostly related to some potential biases associated with the dataset assessed here. It is expected that a number of outbreaks remained unreported, and thus, were not assessed here; if reporting bias selectively affected certain regions or periods of time, results may have been affected. Additionally, data on the type (backyard, smallholder, or commercial) of farm where the outbreaks occurred was not available. The different types of farms could be a significant risk factor associated with the ASF outbreak due to varying levels of biosecurity, population density, veterinary services, and awareness of the disease. Additionally, there may be differences in the swine population between smallholder farms and commercial farms, with smallholder farms typically having a population of less than 20 heads. However, it is important to note that most (70.6%) swine farmers in the Philippines are smallholder or backyard farmers, whereas the remaining 29.4% are considered commercial farms [27]. This ratio remained relatively stable during the ASF epidemic, suggesting that the proportion of affected farms may have been similar. However, analysis of information regarding farm types, which was not available to us, would be a valuable asset for conducting further analyses. The National Zoning and Movement plan for African Swine Fever guideline [9], an early government policy implemented in the Philippines, has faced serious challenges for implementation, explained, at least in part, by a combination of political, socio-cultural, and economic factors. Results of the research here may assist the national and local government units in the Philippines to allocate resources effectively in terms of space and time, ultimately contributing to the design and implementation of effective disease monitoring and future repopulation efforts.

In conclusion, our research offers valuable insights into the dynamics of ASF outbreaks over a 3-year period. Our analysis shows a clear seasonal pattern with higher frequencies occurring from August to October and lower frequencies during April to May. In terms of space, disease was more severe in the Northern island of Luzon, particularly during the second half of the year. These spatial and temporal patterns can inform disease prevention and control efforts in the future, and help authorities prepare for and mitigate the impact of potential ASF outbreaks.

## Figures and Tables

**Figure 1 microorganisms-11-01492-f001:**
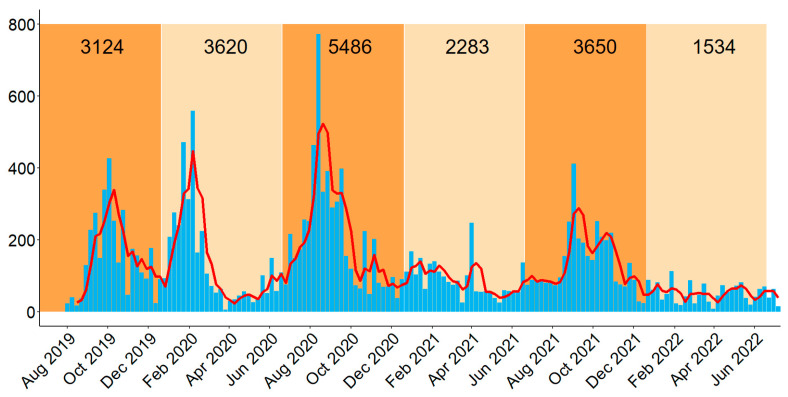
Weekly number of African Swine Fever (ASF) outbreaks (blue bars) and 3-week moving average (red line) reported in the Philippines between 16 August 2019 and 20 July 2022. The numbers in the top indicate the cumulative number of outbreaks reported on the first (1 January–30 June) and second (1 July–31 December) semester of each year, which is indicated by light and dark orange background shades, respectively.

**Figure 2 microorganisms-11-01492-f002:**
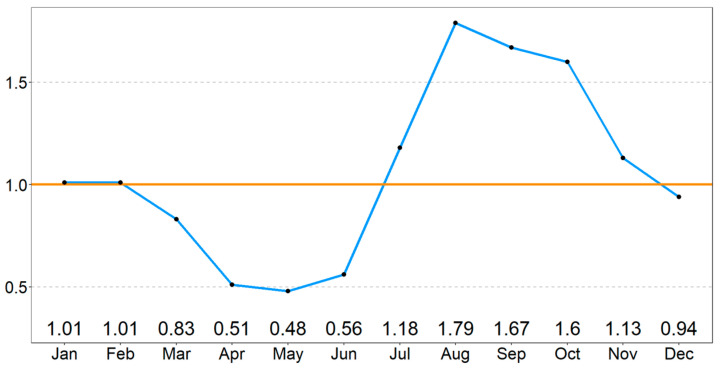
Seasonal index, S*i* (blue line), computed as the ratio between the cumulative number and the monthly cumulative average of African Swine Fever (ASF) outbreaks reported per month in the Philippines between 16 August 2019 and 20 July 2022. The orange line indicates the value of S*i* = 1.

**Figure 3 microorganisms-11-01492-f003:**
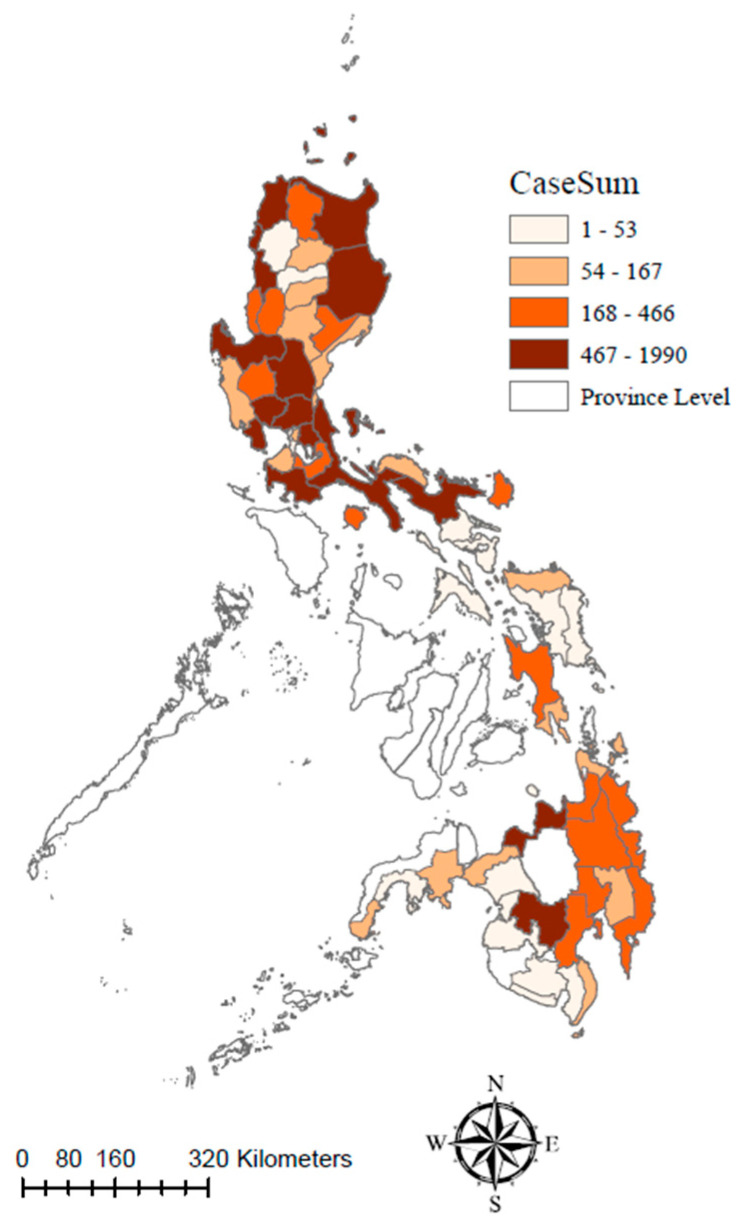
Cumulative number (grouped per quartiles) of African Swine Fever (ASF) outbreaks reported per province in the Philippines between 16 August 2019 and 20 July 2022.

**Figure 4 microorganisms-11-01492-f004:**
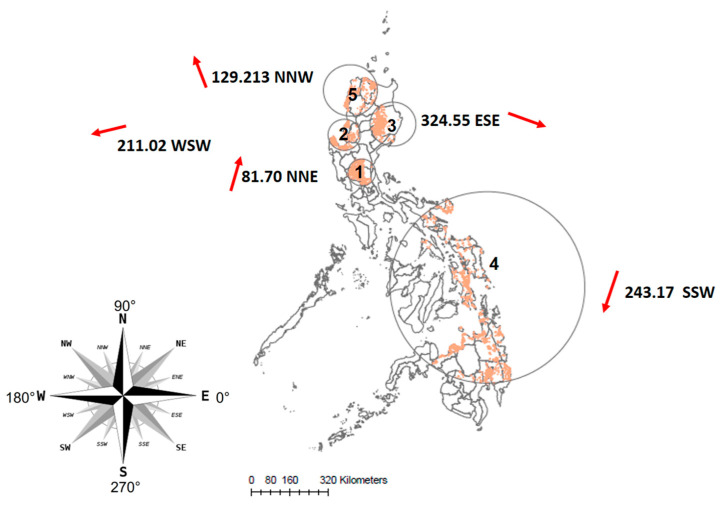
Space–time clusters and corresponding direction (red arrows). Each circle represents a space–time cluster, and the numbers (1 to 5) indicates the order on time axis.

**Table 1 microorganisms-11-01492-t001:** Clusters of African Swine Fever (ASF) outbreaks reported in the Philippines between 16 August 2019 and 20 July 2022, identified using the space–time permutation model of the scan statistics. The results of the directional test for each of the clusters are also shown.

Space–Time Clusters of ASF Epidemic in the Philippines								Directional Test				
Cluster	Radius (km)	Year	Interval	Observed	Expected	ODE	*p* value	Outbreak	Average Angle	Orientation	Concentration	*p*-value
1	55.86	2019	Aug–Dec	2850	633.98	4.50	<0.00001	3124.00	81.70	NNE	0.28	0.001
2	64.90	2020	Mar–Jun	494	89.58	5.51	<0.00001	1202.00	211.02	WSW	0.12	0.001
3	91.67	2020	Aug–Oct	1496	361.67	4.14	<0.00001	3878.00	324.55	ESE	0.04	0.001
4	399.64	2021	Jan–Apr	1261	310.01	4.07	<0.00001	1813.00	243.17	SSW	0.10	0.001
5	107.00	2021	Jul–Oct	1776	295.63	6.01	<0.00001	2721.00	129.21	NNW	0.17	0.001

## Data Availability

Data are the property of the Philippines government and have been shared with us to conduct the research here to support training activities in the country. Any requests should be directed to the Philippine Bureau of Animal Industry and to the International Training Center on Pig Husbandry of the Philippines.

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
