# Peer review of "Space–Time Dynamics of African Swine Fever Spread in the Philippines"

_microorganisms, 2023, doi:10.3390/microorganisms11061492_

Round 1

Reviewer 1 Report

This is an interesting, well written, and potentially very valuable space-time mapping of the African Swine Fever outbreaks in the Philippines.  My major questions are not so much in the conduct of the study, but rather in its interpretation.

1.  For the non-ASF-knowledgeable reader, some basic definitions should be provided.  i) what constitutes an "outbreak"? e.g., is it a single case; does it have to be spatially or temporally separated from another case? how many cases, on average, per outbreak? ii) what constitutes a "cluster"? iii) what does "concentration" in Table 1 refer to?

2.  Although I agree that these kind of data should allow one to better understand the dynamics and facilitate implementation of outbreak-preventing mechanisms, the current analysis seems superficial at best and more reliant on studies in other countries than the authors' own data.  Some specific examples:

a) The authors invoke seasonal fluctuations in rainfall as a probable cause of the Aug/Sep ASF peaks, which is plausible.  But did the rainy seasons actually peak in those two or three years? Also, are there any other factors (e.g., when hogs are slaughtered, perhaps for export markets? or hog breeding cycles?) that coincide with those monthly peaks?

b) The authors note that other countries have correlated holiday-related movement of people and pork products with increased transmission, and that in the Philippines the Christmas season would likely represent a triggering factor.  But wouldn't the decrease in ASF outbreaks starting in February and extending through May suggest this is NOT an important factor in the Philippines, unlike what is stated in lines 313-4?

c) While the assignment of vectors to describe the spread of ASF outbreaks within clusters (if I understood correctly) seems quite elegant, it would help the reader if the authors were to offer possible explanations as to why they wouldn't spread outwardly as concentric circles around the initial cases, rather than in the linear fashion they depict. 

d) Similarly, in lines 290-1, the authors cite "The complex traffic hub, higher human population density as explanations for the patterns," but this is so vague as to be difficult to interpret.

3.  Lines 227-8 don't belong in the Text as they are notes to the Editor.

4.  Finally, I think the introduction would benefit from being shortened--my guess is that the Philippine-specific section (lines 50-109) could be cut almost in half with no loss in content and an increase in readability.

Author Response

Response to Reviewer 1 Comments

This is an interesting, well written, and potentially very valuable space-time mapping of the African Swine Fever outbreaks in the Philippines.  My major questions are not so much in the conduct of the study, but rather in its interpretation.

  1. For the non-ASF-knowledgeable reader, some basic definitions should be provided.  i) what constitutes an "outbreak"? e.g., is it a single case; does it have to be spatially or temporally separated from another case? how many cases, on average, per outbreak? ii) what constitutes a "cluster"? iii) what does "concentration" in Table 1 refer to?

(i) The definition of outbreaks was provided in lines 110-114. As our data is at the farm-level, the exact number of cases per outbreak is unavailable. The accuracy of estimates on the number of individual cases per outbreak would be considered only of relative relevance given that the number of cases identified is heavily dependent on the specific time when the observation within the outbreak was made. For that reason, the information may be misleading –and in any case, was not made available to us.

(ii) The definition of cluster has been provided in lines 135-137.

(iii) A detailed description of concentration has been provided in the method section, specifically in lines 176-179.

  1. Although I agree that these kind of data should allow one to better understand the dynamics and facilitate implementation of outbreak-preventing mechanisms, the current analysis seems superficial at best and more reliant on studies in other countries than the authors' own data.  Some specific examples:
  2. a) The authors invoke seasonal fluctuations in rainfall as a probable cause of the Aug/Sep ASF peaks, which is plausible.  But did the rainy seasons actually peak in those two or three years? Also, are there any other factors (e.g., when hogs are slaughtered, perhaps for export markets? or hog breeding cycles?) that coincide with those monthly peaks?
  3. b) The authors note that other countries have correlated holiday-related movement of people and pork products with increased transmission, and that in the Philippines the Christmas season would likely represent a triggering factor.  But wouldn't the decreasein ASF outbreaks starting in February and extending through May suggest this is NOT an important factor in the Philippines, unlike what is stated in lines 313-4?

Following the reviewer comments, which were appropriate, we decided to organize a workshop with individuals with field experience with ASF in the Philippines, to contrast our results and gather their opinion about interpretation. This activity may be considered a sort of “validation”, although we refrained to use the term validation in the manuscript because of the negative connotations it may have for some readers. Workshop participants provided information that was consistent with our initial interpretation and added additional interpretations that were subsequently included in the manuscript. The methods, results, and discussion sections were expanded in lines 183-193, 223-232, 271-274 and 319-327 to include this information.

  1. c) While the assignment of vectors to describe the spread of ASF outbreaks within clusters (if I understood correctly) seems quite elegant, it would help the reader if the authors were to offer possible explanations as to why they wouldn't spread outwardly as concentric circles around the initial cases, rather than in the linear fashion they depict. 

As the reviewer has indicated, the pattern of spread is of interest to understand the nature of local spread. Concentric circles, compatible with diffusion, would have resulted in no specific direction of spread. However, the significant concentrations of spread identified in certain directions suggests that there were specific factors (such as road movements, for instance) that contributed to the spread.

d) Similarly, in lines 290-1, the authors cite "The complex traffic hub, higher human population density as explanations for the patterns," but this is so vague as to be difficult to interpret.

The sentence has been re-written for clarity in lines 300-305.

  1. Lines 227-8 don't belong in the Text as they are notes to the Editor.

The lines have been deleted as suggested by the reviewer.

  1. Finally, I think the introduction would benefit from being shortened--my guess is that the Philippine-specific section (lines 50-109) could be cut almost in half with no loss in content and an increase in readability.

Thank you for the recommendation, and we have significantly shortened the introduction, reducing it to almost half of its original length. Line 50-82.

Reviewer 2 Report

good morning

I adde some comments, according to row number in article proposal:

30:  data exist relative to poor or lack of biosecurity in East EU/Russian pig farms, including backyard, as involved in ASF sperad; I would consider adding this issue.

318-322: this is really a limiting  factor; lack of this kind of information may affect the practical significance of previous data/results, as stated in row 327.

I would really encourage the Authors to extend their analysis including the data as mentioned in rows 322-327.

I am anyway aware that most probably it will not be possible to add/integrate these data at this stage

Therefore, it could be taken into account  proposing a new article (a sort oif "part II") which could integrate the missing data, and it would result  of strong interest, help and support to farmers and practitioners (e.g relative to biosecurity) and decision-makers (e.g relative to movement, goods restrictions, etc) and therefore give a practical application in field for the results of this interesting analysis. 

Author Response

Response to Reviewer 2 Comments

good morning

I adde some comments, according to row number in article proposal:

30:  data exist relative to poor or lack of biosecurity in East EU/Russian pig farms, including backyard, as involved in ASF sperad; I would consider adding this issue.

We have included additional references that shed light on the outbreak of African swine fever (ASF) in Russian pig farms and its connection to backyard farming practices in lines 41-42.

318-322: this is really a limiting factor; lack of this kind of information may affect the practical significance of previous data/results, as stated in row 327.

I would really encourage the Authors to extend their analysis including the data as mentioned in rows 322-327.

I am anyway aware that most probably it will not be possible to add/integrate these data at this stage

While it presents a challenge to determine the precise livestock population and identify specific farming types in our dataset, we have incorporated data from the Philippines Statistics Authority as a point of reference. This data highlights that, considering the ratio of approximately 30% commercial farms to 70% backyard farms, we can expect that the majority of outbreaks in most regions consist of fewer than 20 cases. Further elaboration on this matter can be found in lines 337-342.

Therefore, it could be taken into account  proposing a new article (a sort oif "part II") which could integrate the missing data, and it would result  of strong interest, help and support to farmers and practitioners (e.g relative to biosecurity) and decision-makers (e.g relative to movement, goods restrictions, etc) and therefore give a practical application in field for the results of this interesting analysis. 

We totally agree with the suggestion from reviewer 2. Considering the endemic status of ASF in the Philippines, we are collaborating with the ITCPH and the Government of Batangas, a province known for its significant swine production over the country. The objective is to strengthen passive/ enhanced passive and active surveillance efforts and to evaluate the effectiveness of point-of-care diagnostic methods across various farming systems.

Round 2

Reviewer 1 Report

The authors have adequately addressed my concerns--and emerged with an interesting and plausible hypothesis.